# Year-round stratospheric aerosol backscatter ratios calculated from lidar measurements above Northern Norway

Arvid Langenbach[1], Gerd Baumgarten[1], Jens Fiedler[1], Franz-Josef Lübken[1], Christian von Savigny[2], and Jacob Zalach[2]

[1]Leibniz Institut für Atmoshärenphysik an der Universität Rostock, Schlossstraße 6, 18225 Kühlungsborn
[2]Universität Greifswald, Felix-Hausdorff-Str. 6, 17489 Greifswald

*Correspondence to:* Arvid Langenbach (langenbach@iap-kborn.de)

**Abstract.** We present a new method for calculating backscatter ratios of the stratospheric sulphate aerosol (SSA) layer from day- and nighttime lidar measurements. Using this new method we show a first year-round dataset of stratospheric aerosol backscatter ratios at high latitudes. The SSA layer is located at altitudes between the tropopause and about 30 km. It is of fundamental importance for the radiative balance of the atmosphere. We use a state-of-the-art Rayleigh-Mie-Raman lidar at the Arctic Lidar Observatory for Middle Atmosphere Research (ALOMAR) station located in Northern Norway (69° N, 16° E, 380 m a.s.l.). For nighttime measurements the aerosol backscatter ratios are derived using elastic and inelastic backscatter of the emitted laser wavelengths 355, 532 and 1064 nm. The set-up of the lidar allows to perform measurements with a resolution of about 5 minutes in time and 150 m in altitude with high quality, allowing to identify multiple sub-layers in the stratospheric aerosol layer of less than 1 km vertical thickness.

We introduce a method to extend the dataset throughout the summer when measurements need to be performed under permanent daytime conditions. For that purpose we approximate the backscatter ratios from color ratios of elastic scattering and apply a correction function. We calculate the correction function using the average backscatter ratio profile at 355 nm from about 1700 hours of nighttime measurements from the years 2000 to 2018. Using the new method we finally present a year-round dataset based on about 4100 hours of measurements during the years 2014 to 2017.

*Copyright statement.* TEXT

## 1 Introduction

The importance of stratospheric sulphate aerosol (SSA) for the radiative balance and the ozone chemistry of the atmosphere is widely accepted. Long-term observations of the stratospheric aerosol layer are crucial for the analysis of global atmospheric temperature and ozone layer variability (Tho, 2006; Solomon et al., 2011). First in situ measurements of SSA have been performed by Christian Junge and co-workers (Junge and Manson, 1961). They found a distinct layer between 15 and 25 km altitude with a peak at 20 km (Junge et al., 1961a, b). The stratospheric aerosol layer is therefore often referred to as Junge-layer. Remote sensing of the aerosol layer by lidar was started by Bartusek and Gambling (1971). Global satellite observations

of SSA began in the late 1970s (e.g. Tho, 2006; Kremser et al., 2016). The upper boundary of the SSA layer is determined by the evaporation of the aerosol particles due to rising temperatures in the stratosphere as well as sedimentation (Hofmann et al., 1985). The tropopause is the base of the aerosol layer since the upper tropospheric aerosol loads are often much lower than in the stratosphere (Kremser et al., 2016).

Understanding the formation and life-cycle of SSA are impossible without understanding the processes controlling sulfur in the stratosphere. Stratospheric sulfur can be found in a broad variety of molecules, such as carbon disulfide ($CS_2$), sulfur dioxide ($SO_2$), carbonyl sulfide (OCS) and sulfuric acid ($H_2SO_4$) (English et al., 2011). SSA typically consist of 75 % sulfuric acid/water ($H_2SO_4-H_2O$) solution droplets (Tho, 2006). In volcanically quiescent periods, the main source for these droplets are $CS_2$ and OCS, which are emitted at the Earth's surface and lifted into the stratosphere by deep convection and the Brewer-
Dobson circulation (Khaykin et al., 2017). They then react in multiple steps via $SO_2$ into sulfuric acid (Kremser et al., 2016). Stratospheric aerosols are primarily washed out by sedimentation and through the quasi-isentropic transport of air masses in tropopause folds (Tho, 2006).

Moreover, the SSA variability is dominated by major volcanic eruptions injecting sulfur directly into the stratosphere. These episodic but powerful eruptions can overlay the permanent stratospheric aerosol layer (referred to as 'background' aerosol)
for years and have a global cooling effect on the surface in the order of a few tenths of a degree Celsius (Robock and Mao, 1995). The fact that aerosols from large volcanic eruptions have global effects was first determined by worldwide observations of optical phenomena following the eruption of Krakatoa in 1883 (Symons, 1888). After the Mount Pinatubo eruption in 1991 the stratospheric sulfur burden was increased by a factor of 60 above background levels and remained elevated by a factor of 10 well into 1993 (McCormick et al., 1995).

The long-term development of SSA has been discussed in various studies (Kremser et al., 2016). Ignoring periods with volcanically enhanced SSA, observations covering the time span between 1970 and 2004 did not show significant changes in the background aerosol (Deshler et al., 2006). Newer studies show rising levels of SSA since 2002 (Hofmann et al., 2009; Vernier et al., 2011; Trickl et al., 2013; von Savigny et al., 2015). The reason for this increase is being debated. Originally the rise of the aerosol levels was connected to a fast increase in Asian sulfur emissions (Hofmann et al., 2009). More recent studies
show an increase in non volcanic aerosol inside of the Asian Tropopause Aerosol Layer. This layer occurs during the northern summer above the Asian monsoon (Vernier et al., 2015; Yu et al., 2015). Vernier et al. (2011) showed with the help of global satellite observations that weaker eruptions also influence the stratospheric aerosol layer. These moderate eruptions are much less powerful than El Chichón or Pinatubo and the effect on stratospheric aerosol levels are much smaller. However, several studies have shown that they have an impact on global surface temperatures nevertheless (Solomon et al., 2011; Fyfe et al.,
2013; Santer et al., 2014, 2015; Andersson et al., 2015).

Accurate long-term measurements are essential to quantify background, volcanic and anthropogenic changes in the strato-spheric aerosol layer. While there have been several reports on seasonal and decadal scale ground based lidar measurements of the aerosol layer at middle latitudes (Trickl et al., 2013; Khaykin et al., 2017; Zuev et al., 2017) there are no year-round or multi year measurements of the stratospheric aerosol layer at high latitudes.

The main goal of this study is to present a year-round stratospheric aerosol record at polar latitudes for the first time, applying elastic laser scattering at three different wavelengths including measurements under full daylight conditions. This study introduces a method to approximate backscatter ratios of the stratospheric aerosol based on measurements of color ratios including a quantification of uncertainties. For this we use elastic and inelastic scattering measured during nighttime in the years 2000 to 2018. To show the performance of the new method we focus on a year-round dataset accumulated between 2014 to 2017. In section 2 the instrumental set-up and the data processing is described. Section 3 summarizes the extension of the dataset using measurements performed during permanent daylight in summer. In section 4 we apply this new method to the years 2014 to 2017 and present a year-round climatology of SSA backscatter ratios.

## 2 Instrument and Method

The Rayleigh-Mie-Raman lidar used in this study is installed at the Arctic Lidar Observatory for Middle Atmosphere Research (ALOMAR) in Northern Norway (69.3° N, 16.0° E, 380 m a.s.l.). The lidar is employed for investigating the Arctic middle atmosphere in the 15 to 90 km altitude range (von Zahn et al., 2000). The instrument is optimized to measure atmospheric temperatures, winds and aerosols (Fiedler et al., 2008; Baumgarten, 2010). The lidar uses two power lasers and two receiving telescopes. Each of the two Nd:YAG power lasers generates 30 pulses per second with a pulse energy of 165, 500, and 465 mJ at the wavelengths of 1064, 532, 355 nm, respectively. The telescopes have a diameter of 1.8 m and are tiltable from zenith pointing to 30° off-zenith to perform wind measurements and observations in a common volume with sounding rockets (Baumgarten et al., 2002).

The light collected by the telescopes is coupled alternatingly into the detection system synchronized to the alternatingly firing lasers. Fig. 1 shows a schematic overview of the detection system, limited to the channels used in this study. The light is separated according to wavelength using dichroic beam splitters. The detection system is capable of detecting backscattered light at seven wavelengths during night-time (elastic: 355, 532, 1064 nm; inelastic: 387, 529, 530, 608 nm). During daytime the system detects backscattered light at three elastic wavelengths (355, 532, 1064 nm) using additional Fabry-Pérot etalon based filters to reduce the solar background (von Zahn et al., 2000). The lidar measurements have been used before for calculating particle properties in the mesosphere and the upper stratosphere (e.g. von Cossart et al., 1999; Baumgarten et al., 2008; Gerding et al., 2003). For the study of aerosols in the stratosphere, the channels were extended with intensity cascaded detectors to allow for simultaneous measurements from the troposphere to the mesosphere. The backscatter signal is recorded with a time and range resolution of 30 seconds and 50 m, respectively.

### 2.1 Processing of the raw data

Before we start the actual aerosol retrieval we perform the following steps:

- Dead time correction

  Once a photon is detected, a minimum time span has to pass before another photon can be detected. This dead time $\tau$ is about 20 to 50 ns for the detectors used in this study. The corrected number of counted photons $N$ is calculated from the

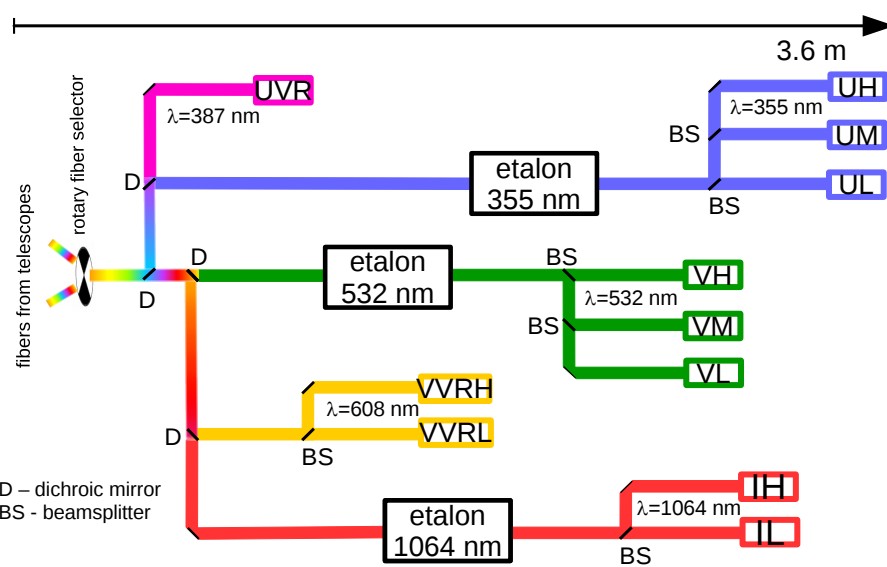

**Figure 1.** Simplified overview of the detection system of the lidar. The light collected by the two telescopes enters the detection system through a fiber selector (left) that is synchronized to the lasers. It is then separated according to wavelength with dichroic mirrors (D) and according to intensity by beamsplitters (BS). In daylight conditions, the light is guided through etalons to suppress solar background. At the end of each detection branch, the photons are converted to electrical pulses using avalanche photo diodes and photomultiplier tubes. The names of these detectors are formed as follows: first letter: spectral range (U-ultraviolet, V-visible, I-infrared), last letter: sensitivity (L-low, M-middle, H-high). Middle letters "VR" indicate vibrational Raman scattered light.

dead time and the count rate $N_c$ (Kovalev and Eichinger, 2004):

$$N = \frac{N_c}{1 - \tau \cdot N_c} \tag{1}$$

- Background subtraction

  The telescopes also collect light from scattered solar photons, stars or airglow. The mean signal from above 100 km rep-
5  resents this background signal since backscattered laser light from these altitudes is negligible. This mean is subtracted from the signals at lower altitudes.

- Gridding of lidar data:

  The raw data is averaged in time for 5 minutes and in altitude to bins of 150 m taking into account the different pointing angles of the telescopes. All altitudes in this work are referenced to the mean sea level.

10 - Correction for extinction by Rayleigh scattering

  The intensity of the outgoing laser beam decreases slightly with altitude since a small fraction of the laser light is scattered by air molecules and aerosols. This also reduces the scattered, downward propagating light. The magnitude of

this effect depends on the wavelength and the density of the atmosphere (Penndorf, 1957). We use air densities from a numerical weather prediction model (see below).

- Correction for extinction by ozone

  Some part of the laser light is absorbed by $O_3$, in particular in the Chappuis bands affecting the laser wavelengths 532 and 608 nm. This effect is corrected using the $O_3$ absorption cross sections from Gorshelev et al. (2014) and the $O_3$ mixing ratios and air density from a climatological model (see below).

- Combination of intensity cascaded detectors

  We combine the signals of intensity cascaded detector groups by normalizing the lower intensity signal to the higher intensity signal in an altitude range where both detectors provide a sufficient signal $S$ with a relative uncertainty of better than $\Delta S/S < 0.1$.

The measurement uncertainty $\Delta S$ is calculated from the initial count of photons assuming a Poisson distribution. This uncertainty of the raw counts is then propagated through the processing steps listed above.

We calculate the Rayleigh- and ozone-extinction using densities and ozone mixing ratios provided by the European Centre for Medium-Range Weather Forecasts (ECMWF). The data from the Integrated Forecasting System of ECMWF is extracted for the location of ALOMAR on hourly basis. The model data is then interpolated to the lidar time and altitude bins (5 min and 150 m). We performed a sensitivity study using air densities from an empirical model (MSISE-00; Picone et al. (2002)) and a mean seasonal cycle for ozone (Rosenlof et al., 2015). It turned out that this leads to unrealistic backscatter ratios as the actual ozone profile significantly deviates from the climatological mean, especially in winter (see discussion of ozone extinction in section 3).

The different detector groups and their corresponding scattering mechanisms are summarized in table 1. An example for the signals of individual detectors and the combined signals $S^\lambda$ is shown in Fig. 2. The data is averaged for 17 hours starting at 13 UT on January 27, 2018. About 7 hours were performed under daytime conditions while 10 hours of the measurement were performed under nighttime conditions. The telescopes were pointing for about 1.5 hours to zenith and for the rest of the measurement $20°$ off zenith towards North and East. For plotting the data we have calculated the mean signals of the two telescopes. For the elastic scattered signals we observe a sudden increase in the signals below 30 km which is caused by tropospheric (below about 10 km) and polar stratospheric clouds ($\sim$15 to 25 km).

## 2.2 Calculation of backscatter ratios

The standard method to characterize the aerosol content in the atmosphere from lidar signals is to calculate the backscatter ratio $R$ from the molecule and aerosol backscatter coefficients $\beta_m$ and $\beta_a$, respectively (see, (Fernald, 1984; Klett, 1985; Ansmann et al., 1990, e.g.)):

$$R = \frac{\beta_m + \beta_a}{\beta_m} \tag{2}$$

**Table 1.** Labels of combined signals and individual detectors. The name indicates in the first letter the spectral range (U-ultraviolet, V-visible, I-infrared), and with the last letter the sensitivity (L-low, M-middle, H-high). The letter pair "VR" indicates vibrational Raman scattered light. Indices represent the particular wavelength.

| combined signal | detector | scattering process | daylight capability |
|---|---|---|---|
| $S^{355}$ | $UL_{355}$ | elastic (Rayleigh and Mie) | yes |
| | $UM_{355}$ | | |
| | $UH_{355}$ | | |
| $S^{532}$ | $VL_{532}$ | elastic (Rayleigh and Mie) | yes |
| | $VM_{532}$ | | |
| | $VH_{532}$ | | |
| $S^{1064}$ | $IL_{1064}$ | elastic (Rayleigh and Mie) | yes |
| | $IH_{1064}$ | | |
| $S^{387}$ | $UVR_{387}$ | inelastic (Raman) | no |
| $S^{608}$ | $VVRL_{608}$ | inelastic (Raman) | no |
| | $VVRH_{608}$ | | |

In our case the backscatter coefficients are proportional to the corrected signals $S$, shown in Fig. 2c. Let's for example consider scattering at $\lambda = 1064$ nm:

$$R^{1064} = \frac{S_m^{1064} + S_a^{1064}}{S_m^{1064}} = \frac{S^{1064}}{S_m^{1064}} \tag{3}$$

The challenge is to retrieve the signal scattered by molecules only ($S_m^{1064}$) since the signal received by the lidar ($S^{1064}$) contains both contributions from scattering on molecules and aerosols. For this we use the signal from Raman backscattering on $N_2$ at $\lambda = 387$ nm which contains molecular scattering only: $S^{387}$. Since Raman scattering is less efficient compared to Rayleigh and Mie scattering, $S^{387}$ is much smaller compared to $S_m^{1064}$ at any given altitude. However, using the correction and applying the processing steps in section 2.1 both signals are proportional to each other ($S^{387} \propto S_m^{1064}$) since they are both given by molecular scattering. We determine the proportionality constant $F$ at an altitude $z_F$ where no aerosols exist, hence $S^{1064}$ equals $S_m^{1064}$, which is typically the case above 34 km:

$$F_{387}^{1064} = \left\langle \frac{S^{1064}}{S^{387}} \right\rangle_{z=z_F} \tag{4}$$

This allows to derive $S_m^{1064}$ at any given altitude:

$$S_m^{1064} = F_{387}^{1064} \cdot S^{387} \tag{5}$$

therefore Eq. 3 leads to:

$$R_{387}^{1064} = \frac{S^{1064}}{F_{387}^{1064} \cdot S^{387}} \tag{6}$$

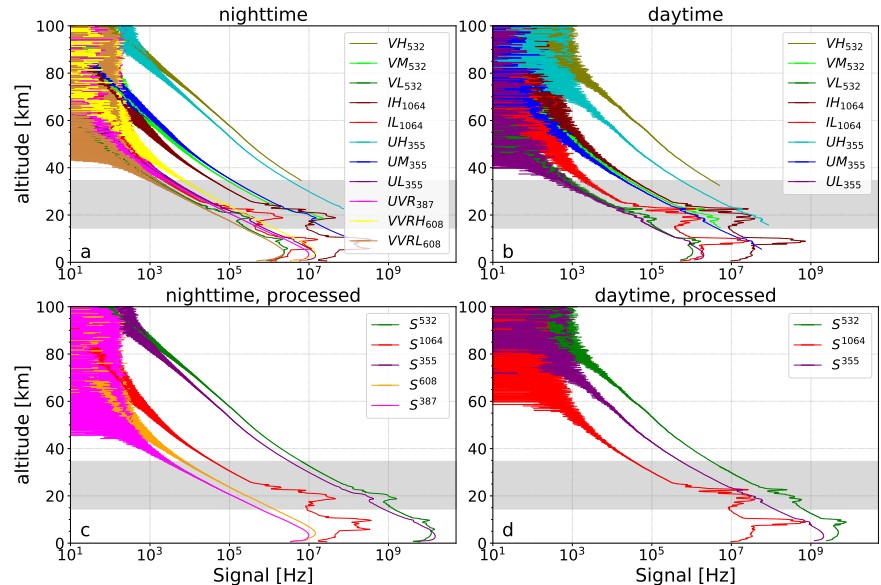

**Figure 2.** Time averaged altitude profiles of backscattered signals for a 17 hour long measurement starting at 13 UT on 27 January 2018. (a) and (b) show the count rates of individual detectors, (c) and (d) the resulting profiles after combining the signals of the different detector groups. (a) and (c) show about 10 hours of nighttime measurements and (b) and (d) about 7 hours of daytime measurements. The approximate altitude range of the stratospheric aerosol layer is shown in gray.

This ratio is named $R^{1064}_{387}$ to indicate the wavelength of the elastic backscattered signal ($\lambda = 1064$ nm) and the Raman wavelength ($\lambda = 387$ nm). We use the altitude range $z_F$ from 34 to 38 km to determine $F^{1064}_{387}$ since an initial processing of the dataset showed that the aerosol layer sometimes reaches up to about 34 km. In addition to this rather high normalization altitude we use a two-step process to reduce the effect of an aerosol layer reaching partly up into the normalization altitude range. First, we calculate the mean signal ratio in the normalization range, then we limit the data in the normalization range to those where the signal ratio is within one standard deviation of the mean. More details on this are discussed in section 4.

Additionally to $\lambda = 1064$ nm we also investigated the other two emitted wavelengths, namely $\lambda = 532$ nm and $\lambda = 355$ nm. In total we derive three backscatter ratios: $R^{1064}_{387}, R^{532}_{387}, R^{355}_{387}$, all as a function of altitude. Instead of the Raman signal at $\lambda = 387$ nm we have tentatively used another Raman signal, namely at a wavelength of $\lambda = 608$ nm. It turned out, however, that this is not practical since this signal is partly absorbed by ozone (see Sec. 3). In this work, we focus on $R^{1064}_{387}$ i.e. $\lambda = 1064$ nm as the Rayleigh and Mie scattered signal and $\lambda = 387$ nm as the Raman signal. This combination is superior to the others as ozone extinction does not affect these two wavelengths and the backscatter ratio is found to be largest at $\lambda = 1064$ nm.

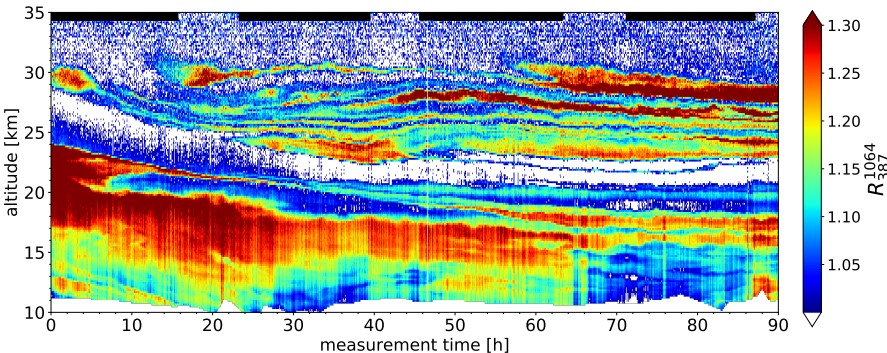

**Figure 3.** Stratospheric aerosol backscatter ratio ($R_{387}^{1064}$) for a measurement starting at 15:30 UT on 18 February 2018. The time resolution is 5 minutes and the altitude resolution is 150 m. Black bars at the top indicate nighttime configuration. At the bottom end of about 11 km the data at altitudes below the tropopause is masked (white).

## 2.3   Identification of the stratospheric aerosol layer

Only backscatter ratios from altitudes above the tropopause are analysed in order to limit the data to the stratospheric aerosol layer. We calculate the dynamical and the thermal tropopause from ECMWF model data and select the higher value of the two as lower altitude limit for the backscatter ratio profiles.

We also remove measurements that show the presence of polar stratospheric clouds (PSC) (Peter, 1997). From December to February these clouds occur frequently at our location. The calculated backscatter ratio of PSCs is about one order of magnitude larger than that of the background aerosol (e.g. Fig. 2), so we use a simple threshold of $R_{387}^{1064} > 2.0$ to exclude PSC from our dataset of stratospheric aerosols.

     An example for a backscatter ratio $R_{387}^{1064}$ of the stratospheric aerosol layer from 88 hours of measurement starting on
February 18, 2018 is shown in Fig. 3. We observe a highly dynamic stratospheric aerosol layer consisting of multiple sub-layers. There are several layers thinner than one km remaining separated and partially moving in parallel over several days. It should be emphasized that these layers are not connected to PSCs for two reasons: (1) The maximum backscatter ratio is well below the PSC threshold of $R_{387}^{1064} = 2.0$ and (2) the temperatures at the altitudes of the layers were above 210 K and therefore about 15 K above PSC formation temperature (Beyerle and Neuber, 1994).

Fig. 3 shows the evolution of the stratospheric aerosol layer during night and day. As the signal $S^{387}$ (needed to calculate $R_{387}^{1064}$) is not measured during daytime we have calculated the mean signal $S^{387}$ during the night measurements (indicated in Fig. 3) and used this mean profile to calculate $R_{387}^{1064}$. Fig. 3 shows that this method of calculating the backscatter ratio during daytime (using a nearby nighttime measurement) results in a reasonable evolution of the backscatter ratio.

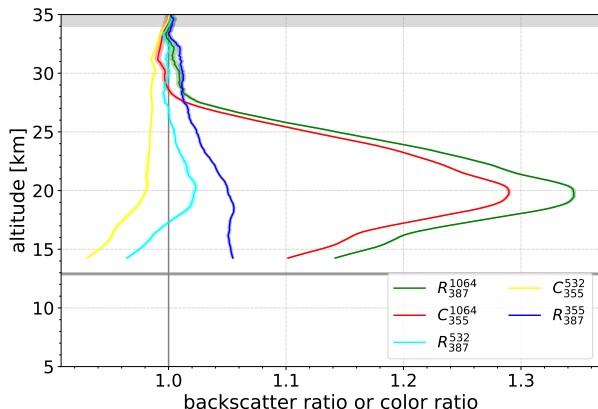

**Figure 4.** Backscatter ratio and color ratio profiles for a 97 hour long measurement starting at 07:45 UT on 10 October 2015. About 46 hours were measured with daytime configuration and 51 hours with nighttime configuration. The gray area at the top indicates part of the normalization altitude. Shaded areas around the lines indicate the measurement uncertainty. The gray line at 13 km indicates the tropopause.

## 3 Calculating the backscatter ratio under daytime conditions

The lidar is situated at 69.3° N, i.e. north of the Polar circle. At this latitude the daytime configuration of the lidar is used from about mid May to mid August. As shown in the previous section the backscatter ratio can be calculated for daytime measurements using a nearby nighttime observation. Unfortunately this is not possible during the summer months due to the permanent daylight. In order to retrieve a solid year-round dataset, we use the year-round available color ratio $C_{355}^{1064}$ with an empirical correction as a proxy for the backscatter ratio $R_{387}^{1064}$.

We define the 'color ratio' $C$, namely the ratio of signals received from two of the three wavelengths (1064 nm, 532 nm, 355 nm) normalized to the signal ratio at an altitude ($z_F$) where no aerosols exist. This is similar to the procedure for calculating the backscatter ratio in section 2.2 and yields, as an example, for 1064 nm and 355 nm:

$$C_{355}^{1064} = \frac{S^{1064}}{F_{355}^{1064} \cdot S^{355}} \tag{7}$$

It is worth noting that the definition of a color ratio used here is different from that used in von Cossart et al. (1999) and Baumgarten et al. (2008).

Fig. 4 shows the mean backscatter ratio and color ratio profiles for a typical measurement in October 2015. The measurement lasted for 97 hours, thereof 46 hours with daytime configuration and 51 hours with nighttime configuration. Here, the backscatter ratio for the daytime part of the measurement is calculated by using the mean signal $S^{387}$ during nighttime observation as described in Sec. 2.3. Comparing $R_{387}^{1064}$ and $C_{355}^{1064}$ we see that $C_{355}^{1064}$ has nearly the same vertical structure but is about 5 % lower than $R_{387}^{1064}$. The difference between $R_{387}^{1064}$ and $C_{355}^{1064}$ is primarily due to the aerosol backscatter signal $S_a^{355}$ included in the signal $S^{355}$. In other words, the color ratio $C_{355}^{1064}$ is a proxy for $R_{387}^{1064}$ that deviates by less than about 5 % from the

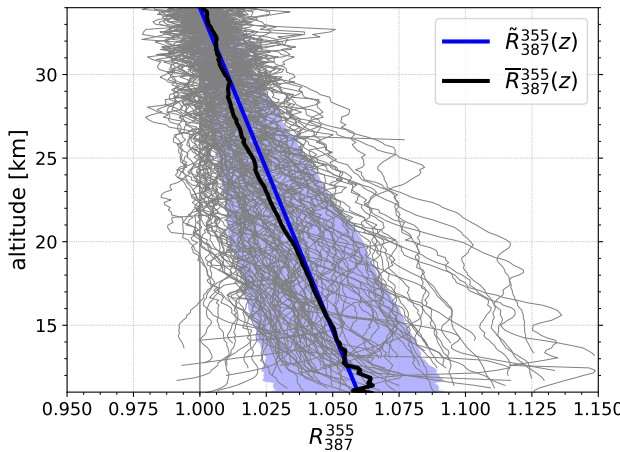

**Figure 5.** Backscatter ratios $R_{387}^{355}$ from nighttime measurements in the period 2000 to 2018. Each gray line represents the mean of the measurement run. The total measurement time is 1789 hours. $\overline{R}_{387}^{355}(z)$ is the mean of all profiles and $\tilde{R}_{387}^{355}(z)$ a linear fit to $\overline{R}_{387}^{355}(z)$. The shaded area represents the standard deviation of the $R_{387}^{355}$ profiles.

true value of $R_{387}^{1064}$ at the peak of the stratospheric aerosol layer. In the following we describe a method how to calculate the backscatter ratio at 1064 nm with respect to the molecular signal at 355 nm ($R_{355}^{1064}$):

$$R_{355}^{1064} = \frac{S^{1064}}{F_{355}^{1064} S_m^{355}} \tag{8}$$

This can be rewritten using the color ratio $C_{355}^{1064}$ and the backscatter ratio $R_{387}^{355}$:

$$5 \quad R_{355}^{1064} \;=\; \frac{S^{1064}}{F_{355}^{1064} F_{387}^{355} S^{387}} \tag{9}$$

$$=\; \frac{S^{1064}}{F_{355}^{1064} S^{355}} \cdot \frac{S^{355}}{F_{387}^{355} S^{387}} \tag{10}$$

$$=\; C_{355}^{1064} \cdot R_{387}^{355} \tag{11}$$

Since the backscatter ratio $R_{387}^{355}$ is not available for daytime measurements we approximate the actual profile by a mean of all available measurements, to calculate an approximated backscatter ratio $R_{355}^{1064}$. Figure 5 shows the mean $R_{387}^{355}$ profile for each

10 of the 103 measurements between 2000 and 2018 which cover a total of 1789 hours. These measurements were selected as they have a relative backscatter ratio uncertainty of less than one percent ($\Delta R_{387}^{355}/R_{387}^{355} < 0.01$).

The profiles of $R_{387}^{355}$ show a linear decrease with altitude towards $R = 1$ at $z = 34$ km. This behaviour is seen very well in the mean profile ($\overline{R}_{387}^{355}(z)$). We make use of this systematic altitude dependence by fitting a linear regression:

$$\tilde{R}_{387}^{355}(z) = \frac{z - 407.95 \text{ km}}{-374.16 \text{ km}} \tag{12}$$

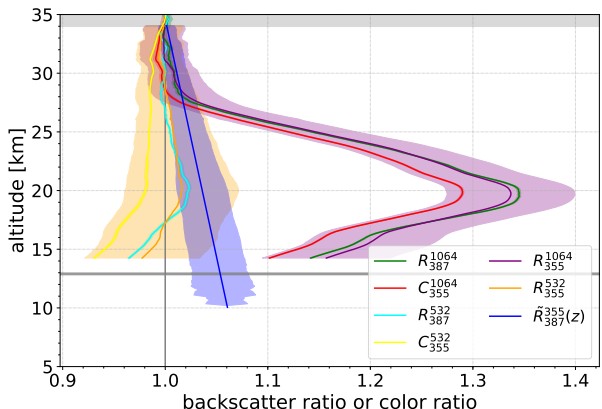

**Figure 6.** Backscatter ratio, color ratio, and approximate backscatter ratio profiles ($R_{355}^{1064}$, $R_{355}^{532}$) for a 97 hour measurement starting 07:45 UT on 10 October 2015, the same measurement as in Fig. 4. Colored shaded areas around the lines indicate the measurement uncertainty. Further explanations are given in Fig. 4.

Finally, we use the fit $\tilde{R}_{387}^{355}(z)$ to calculate the approximated backscatter ratio $R_{355}^{1064}$:

$$R_{355}^{1064} = C_{355}^{1064} \cdot \tilde{R}_{387}^{355}(z) \tag{13}$$

Depending on altitude, the correction has a total effect on the backscatter ratio ranging from 5 % at 15 km to zero at 34 km. Fig. 6 shows the approximate backscatter ratio $R_{355}^{1064}$ for the 97 hour long measurement in October 2015. The approximated backscatter ratio profile $R_{355}^{1064}$ matches the backscatter ratio $R_{387}^{1064}$ by better than 1 % which is well within the calculated uncertainty.

The uncertainty of the corrected profile is dominated by the uncertainty of the linear fit $\tilde{R}_{387}^{355}(z)$. We have calculated the uncertainty of the fit as the standard deviation of the difference between $R_{387}^{1064}$ and $R_{355}^{1064}$ for each altitude. This difference is shown in Fig. 7 for each of the 103 measurements. It is symmetric over the whole altitude range and decreases with altitude. This behavior is as expected, as the effect of the correction tends to zero at 34 km and the $R_{387}^{355}$ profiles for each measurement tend to $R = 1$ at 34 km.

In the same way an approximated backscatter ratio $R_{355}^{532}$ is calculated from the corresponding color ratio $C_{355}^{532}$ and the fit $\tilde{R}_{387}^{355}(z)$:

$$R_{355}^{532} = C_{355}^{532} \cdot \tilde{R}_{387}^{355}(z) \tag{14}$$

Both $R_{387}^{1064}$ and $C_{355}^{1064}$ are not affected by ozone extinction whereas $R_{387}^{532}$ is affected by ozone extinction. We note that $R_{387}^{532}$ is smaller than 1 in limited altitude ranges ($z = 15$ to 17 km). By definition a backscatter ratio should not be smaller than 1 (Eq. 3). This indicates that the true ozone extinction may be different from that used for processing the data since the signal at $\lambda = 532$ nm is stronger affected by ozone extinction than the signal at $\lambda = 355$ nm. Due to the normalization of the backscatter

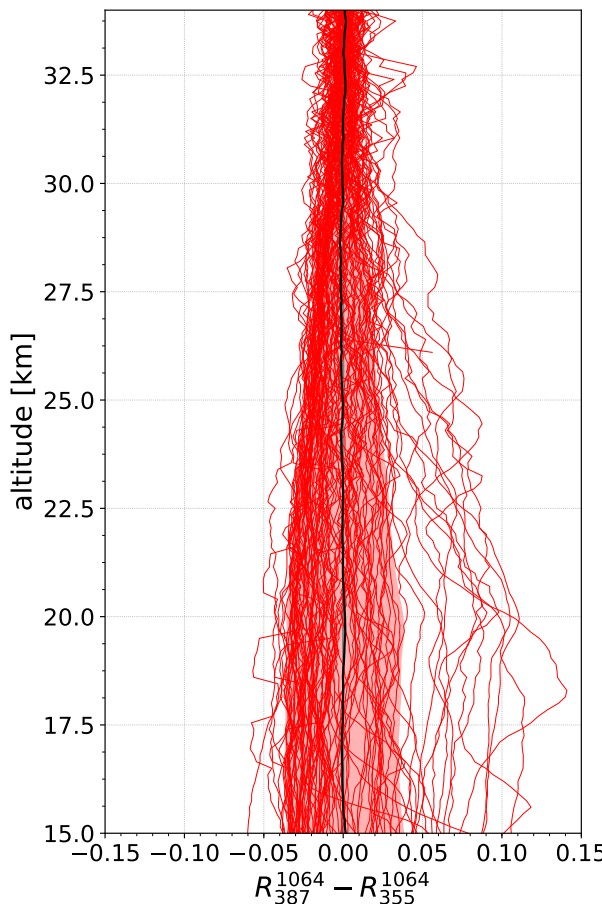

**Figure 7.** Difference of the backscatter ratio $R_{387}^{1064}$ and the approximated backscatter ratio $R_{355}^{1064}$ for 103 measurements between 2000 and 2018 where both ratios are available. The red shadowed area shows the standard deviation of the difference. The black line shows the mean difference.

ratio to 1 in the aerosol free altitude $z_F$ an under-estimation of ozone extinction reduces the backscatter ratio and may result in $R < 1$. A similar effect arises due to a wavelength dependent extinction of the aerosol layer. Here $R$ is reduced at lower altitudes if the wavelength of the elastic backscattered signal is more affected by aerosols than the Raman wavelength (see Eq. 6).

5      The quality of the approximated backscatter ratio $R_{355}^{532}$ is also seen in Fig. 6 as it agrees now well to $R_{387}^{532}$. Notably $R_{355}^{532}$ is larger than 1 in most altitudes (in contrast to $C_{355}^{532}$). We like to point out that approximated backscatter ratios underestimate the true backscatter ratios in cases of strong aerosol loads and overestimate the true backscatter ratios in cases of low aerosol loads since the correction function $\tilde{R}_{387}^{355}(z)$ was derived from measurements with a mean aerosol load.

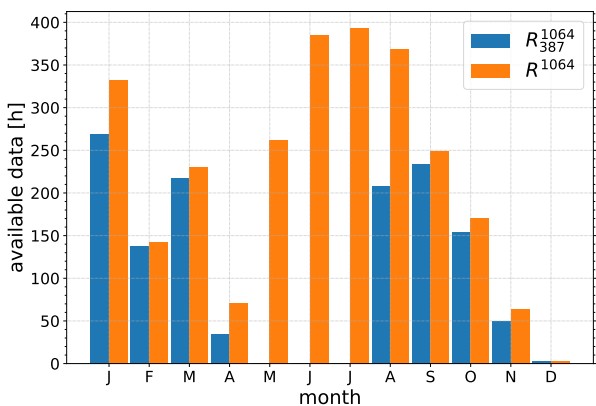

**Figure 8.** Available data in hours for each month for the years 2014 to 2017 for $R_{387}^{1064}$ (blue) and $R^{1064}$ (orange).

Although the backscatter ratios $R_{355}^{1064}$ are only marginally affected by the actual shape of the correction function $\tilde{R}_{387}^{355}(z)$, it is worth discussing the linear decrease of $\tilde{R}_{387}^{355}$ with altitude. First of all, we have not identified an instrumental problem that leads to this linear decrease with altitude; for example an incomplete overlap function would affect both signals $S^{355}$ and $S^{387}$ in the same way. Furthermore ozone extinction can be excluded as a potential error source since it has virtually no impact on these signals.

Using this approach a new dataset $R^{1064}$ is generated consisting of the exact backscatter ratio $R_{387}^{1064}$ for nighttime configuration (or daytime configuration with nearby nighttime measurements) and the approximated backscatter ratios $R_{355}^{1064}$ for the measurements in which only daytime configuration was applied (mostly summertime). The result is a complete year-round dataset of stratospheric aerosol backscatter ratio profiles.

## 4   Results

We have applied the procedure described in Sec. 3 to the data of the years 2014 to 2017. During this period the lidar was operated for 4158 hours and allows to cover a typical seasonal cycle of the stratospheric aerosol. A total of 232 measurement runs were performed. In 24 of those runs PSCs were detected and these runs were therefore excluded. The remaining 208 runs represent 3646 hours of observations. Of these measurements 2391 hours were performed with daytime configuration and 1255 hours with nighttime configuration, respectively. Fig. 8 shows the hours of measurements for the combined dataset $R^{1064}$ and the backscatter ratio $R_{387}^{1064}$. We see that the combined dataset includes the summer months where no measurements of $R_{387}^{1064}$ are available. During the other months the $R^{1064}$ dataset is larger as well, compared to the $R_{387}^{1064}$ dataset.

We calculated the monthly mean backscatter ratios $R_{387}^{1064}$ and $R^{1064}$, omitting the month of December where only 3 hours of measurements are available. For this we first calculated hourly averaged backscatter ratios smoothed in altitude with a running mean of 1.1 km. Then we calculated the average for the two telescopes. Finally the mean of the hourly profiles is calculated

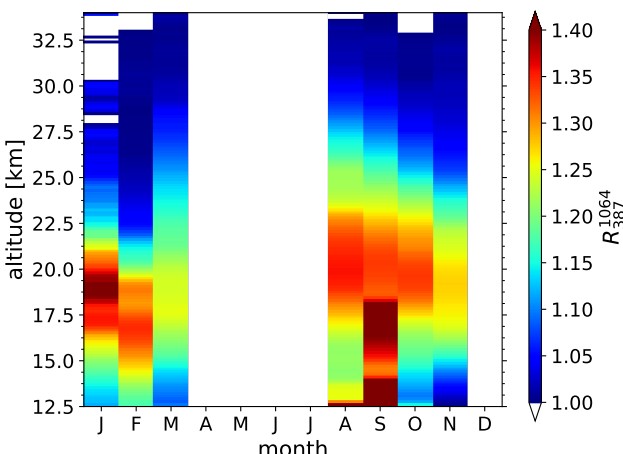

**Figure 9.** Monthly mean aerosol backscatter ratio $R_{387}^{1064}$ from 79 measurement runs from 2014 to 2017. The dataset contains only nighttime measurements and includes a total of 1255 hours.

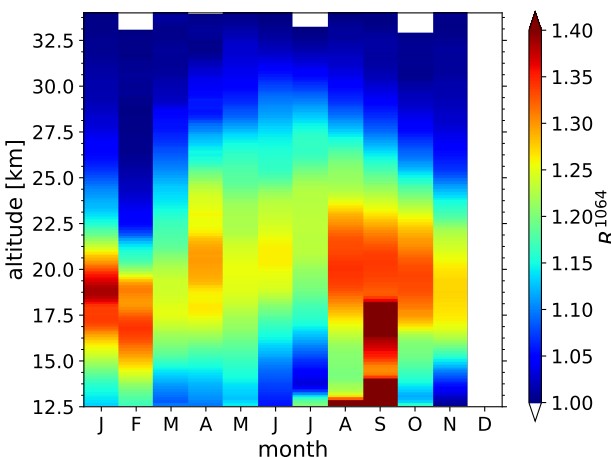

**Figure 10.** Monthly mean aerosol backscatter ratio $R^{1064}$ from 208 measurement runs from 2014 to 2017. The dataset contains night and daytime measurements and includes a total of 3646 hours.

for each month. The standard error of the mean is given by $\sigma_m = \sigma/\sqrt{n}$, where $\sigma$ is the standard deviation and $n$ is the number of measurement hours per month.

The results for the nighttime backscatter ratio $R_{387}^{1064}$ from 79 measurement runs is shown in Fig. 9. We have excluded the month of April in the $R_{387}^{1064}$ dataset as the mean error between the tropopause and 34 km is $\overline{\sigma_m}(R_{387}^{1064}) = 0.025$. This value is
5    rather high when compared to the other months where $\overline{\sigma_m}(R_{387}^{1064})$ is less than 0.01.

The seasonal cycle for the combined dataset $R^{1064}$ from 208 measurement runs is shown in Fig. 10. Now the dataset also covers the summer months. In the months where both datasets $R^{1064}$ and $R_{387}^{1064}$ are available the datasets agree.

The uncertainties of the monthly mean backscatter ratios $R^{1064}$ result to $\sigma_m(R^{1064}) = 0.06$ at 11 km and $\sigma_m(R^{1064}) = 0.02$ at 34 km and are dominated by the uncertainty of the fit $\tilde{R}_{387}^{355}(z)$.

5    The $R^{1064}$ and the $R_{387}^{1064}$ datasets both show enhanced aerosol backscatter in September in the lower stratosphere between 12 km and 18 km. These high aerosol levels originate from wildfires with strong pyrocumulonimbus activity over western Canada in September 2017. The smoke reached Europe 10 days after its injection into the lower stratosphere. This event has been studied in detail by Ansmann et al. (2018).

During the winter months (November to March) the peak and the top of the aerosol layer is located at significantly lower 10 altitudes, compared to the rest of the year. In this period the peak of the layer is most often found below 22 km. This is likely caused by the descending air within the polar vortex in winter. However, the location of the ALOMAR observatory is close to the edge of the polar vortex, so during the winter months we observe sometimes air within the vortex and sometimes outside the polar vortex. This results in a variation of the altitudes of the peak of the layer during the winter months.

A key finding of the new dataset $R^{1064}$ is that the stratospheric aerosol reaches well above 30 km. In summer the typical 15 backscatter ratio at 30 km is $R^{1064} \sim 1.05 \pm 0.02$. So there is clear evidence for aerosols at this altitude. This finding is in contrast to previous studies where the authors assumed an aerosol free altitude starting at 30 km (McCormick et al., 1984; Barnes and Hofmann, 1997; Khaykin et al., 2017; Zuev et al., 2017). However, all of the previous studies were performed at mid to low latitudes.

## 5   Summary and conclusions

20    We have described the calculation of backscatter ratios using elastic (Rayleigh+Mie) and inelastic (Raman) scattering. For investigation of the stratospheric aerosol layer inelastic scattering is available only during nighttime. To our knowledge no lidar instrument exists that measures the stratospheric aerosol during daytime using the classical Raman method of calculating a backscatter ratio from elastic and inelastic scattering.

An extension of the backscatter ratio time series to daytime using close in time nighttime inelastic signals allows to observe 25 small scale structures present in the stratospheric aerosol even during daytime. We present for the first time multiple sharp background aerosol layers of less than 1 km vertical thickness that partly move in parallel to each other over several days.

To calculate the backscatter ratios $R^{1064}$ and $R^{532}$ during daytime where Raman signals are not available we have developed a proxy that is based on the measured color ratio of elastic scattering at the wavelengths 1064, 532, 355 nm and an empirical 30 correction function. The color ratios with respect to $\lambda = 355$ nm already yield reasonable values for the backscatter ratios. However, the color ratios are about 5 % smaller than the backscatter ratios.

A correction function was calculated from multiple measurements of the backscatter ratio $R_{387}^{355}$ in the period from 2000 to 2018. The measurements show a linear decrease of $R_{387}^{355}$ with altitude. The largest uncertainties on the monthly mean

approximated backscatter ratio ($R^{1064}$ and $R^{532}$) are found in the lowest altitudes of the stratospheric aerosol layer, but even there we have observed that the approximated backscatter ratio deviates from the true backscatter ratio by less than 1%.

Using this correction function we calculated for the first time a seasonal cycle of backscatter ratios at high latitudes including the summer months. A dedicated seasonal cycle of the peak of the aerosol layer with higher altitudes in summer and lower altitudes in winter is found. The top altitude of the layer varies in a similar way throughout the year. The aerosol reaches as high as 34 km during the summer months.

We propose to use this new method of calculating the backscatter ratio of the stratospheric aerosol layer from pure elastic scattering for the study of decadal scale variations at high latitudes. Furthermore, the study of variations in stratospheric aerosol at the smallest scales detectable ($dt < 5$ minutes, $dz < 150$ m) will benefit from the method as elastic scattering provides a better signal to noise ratio during night- and daytime.

*Code and data availability.* The datasets used in this study can be obtained by contacting the first author

*Competing interests.* The authors declare that they have no conflict of interest.

*Acknowledgements.* This work benefited from the excellent support of J. Hildebrand, M. Gerding and the dedicated staff at the ALOMAR observatory. The European Centre for Medium-Range Weather Forecasts (ECMWF) is gratefully acknowledged for providing the data. This project is supported by DFG (Deutsche Forschungsgesellschaft, Projektnummer 312991878)

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
