# Peer review of "Year-round stratospheric aerosol backscatter ratios calculated from lidar measurements above Northern Norway"

_Atmospheric Measurement Techniques, 2019_

## Referee Comment (RC1) · Anonymous Referee #1 · 5 Apr 2019

In the paper, "Year-round stratospheric aerosol backscatter ratios calculated from lidar measurements above Northern Norway", the authors present a multiyear stratospheric sulfate aerosol (SSA) dataset from lidar observations at the ALOMAR research station. This paper provides valuable insight into lidar-measured SSA over the Arctic, and the study is appropriate for AMT, however I have a few major concerns with the paper in its current form. These include the overall writing quality of the manuscript and lack of important details of the study. Thus, I recommend a major revision. The authors should address the major and minor comments outlined below for the revised manuscript.

Major comments:

1. Writing quality of manuscript: Many grammatical errors and misspellings are found throughout the text, and acronyms need to be defined. The paper should be thoroughly proofread.

2. Lack of study details: There are several instances in the manuscript that I believe need additional information, as follows:

   a. Page 4, Lines 7-10: Please add more description of ECMWF (e.g., spatial resolution). Why ECMWF? Are there other options? What are the uncertainties associated with the parameters from ECMWF? When you state "converted to a 5 min and 150 m grid", converted from what? Also, add more details on how the Rayleigh and ozone corrections are done.
   b. Sensitivity studies: Please comment on the choices made, and any sensitivity studies completed for normalization altitudes (Page 5, Line 11), wavelengths for elastic/inelastic signals (Page 5, Lines 22-23), and lower limit of data availability (Page 7, Lines 13-14).
   c. Equation 1 (Page 3): Where is this from (reference), or how was it determined?
   d. Page 6 (Section 4): Please add more discussion/explanation for this section, and the importance/purpose of each figure (Figures 4 through 7). For example, why are you showing $R_{355/387}$ for Figure 5 instead of R at other wavelengths? Also, please state why a correction is needed for $CR_{1064/355}$.
   e. Page 7 (Figures 9 and 10): Explain how these figures were created (e.g., averaging), as was included in the figure captions.

3. Conclusions: I believe this entire section needs to be re-worked. Please address the following:

   a. Re-define all acronyms.
   b. I recommend not referencing figures in this section.
   c. Please do not state results that have not been already discussed earlier in the paper. For example, the uncertainties stated in Line 10 of Page 8. This belongs in the Results section.
   d. As mentioned above there are grammatical errors in this section.

e. The narrative does not flow well (including ending with a lone sentence), so I recommend re-writing the entire section.

f. I suggest including bullets or something similar to summarize the main findings of the study.

Minor comments:

1. Page 1, Lines 1-12: Please add a few sentences to the abstract describing the primary results of the study.
2. Page 1, Line 5: Define ALOMAR.
3. Page 1, Line 15: Define SSA. All acronyms should be defined at their first use in the paper.
4. Introduction section: State the location and dates of the study.
5. Page 2, Lines 31-33: lidar measurements of what? R and CR? Explain the parameters of interest. Also, add more motivation as to why this study is important. What is being accomplished/what is the general purpose of this paper?
6. Page 3, Line 5: Add the elevation of the ALOMAR station.
7. Pages 3 and 4 (Section 2): I suggest not using dashes when listing the processing steps. Bullets may work better.
8. Page 4, Line 14: How was this relative uncertainty computed? Please add an explanation to the text.
9. Page 6, Lines 1-2: The layers are also not associated with PSCs because of the PSC screening metrics described on Page 5, correct?
10. Page 7, Line 20: Rephrase "The first picture".
11. Page 7, Line 31: Significantly lower altitudes? Are you comparing 12-18 km to 12-22 km? If so, this sentence does not make sense. This paragraph is confusing, so I recommend revising it.
12. Page 8, Lines 1-4: How do these findings compare with other studies?
13. Figure 2: Add labels, like a-d, to the plots, and refer to them in the caption. How was the altitude range of the stratospheric aerosol layer determined? Also, as a general comment, mention whether the altitudes are referenced to above ground level (AGL) or above mean sea level (AMSL). This should be stated in the text of the paper as well.
14. Figure 4: For the x-axis, I suggest not using a slash symbol (/) here, as this could be confusing. Maybe use "or" instead. Also, the colored shaded areas representing the measurement uncertainties are very difficult to see.
15. Figure 5: Please mention in the caption what the shaded area in blue and black vertical line at $R_{355/387} = 1$ represent.
16. Figure 8 caption: I suggest re-wording "Time of available data".
17. Figures 1-10: I suggest making the text larger for both the axes and color bars.

---

## Referee Comment (RC2) · Anonymous Referee #2 · 12 Apr 2019

This paper presents the analysis of stratospheric aerosols observations using the state of the art Rayleigh-Mie-Raman multiple wavelength lidar at ALOMAR. The stratospheric aerosol layer is observed at 1064 nm with unprecedented high resolution. The topic of the paper is well suitable for publication in AMT. However the description of the data processing and the results should be improved and I recommend a major revision as detailed below.

Equation (1) page 3 for the dead time correction does not seem correct. The correct formulation is: N=Ncount/(1-tau Ncount)

Page 6, lines 10-11, how equation 3 could be applied if the inelastic signal is not

present?

Page 6, lines 15-21, the justification for a linear correction with altitude of the R355/387 is not given. It may hide some instrumental problems in the lidar. This point should be discussed in more details. Also I wonder why the ratios R532/387 and R532/355 on Figure 4 fall below 1 in the lower altitude range. Is it a problem of detector saturation?

Results section page 7 How the standard error of the monthly mean scattering ratio is computed? Is it from the statistical error on lidar signal at different wavelengths? Due to the limited number of available hours of measurements per month and the large variability of the Artcic stratosphere, especially during winter months, the monthly averaged value of the scattering ratio cannot be considered as fully representative of the monthly climatological value for this month.

Page 7, lines 27-29, the increase of aerosol loading in the lower stratosphere in August-September due to smoke from the Canadian fires merits to be discussed in more details than just put in the mean seasonal cycle.

---

## Referee Comment (RC3) · Anonymous Referee #3 · 1 May 2019

The paper is appropriate for AMT, but not in a good shape. Major revisions are needed.

The paper is much too long. Basic lidar stuff is unnecessarily presented in large detail. A compact version is needed.

Introduction:

The importance of the SSA is presented in large detail! Why? One paragraph would be sufficient! On the other hand, one has to read the entire paper to get an idea: What is new here? What is the motivation to write this paper? Figures 9 and 10 tell the reader finally what the step forward is.

Please provide the motivation right in the beginning (second paragraph of the introduction): precise and compact. The shorter the introduction the better.

Good points to be mentioned in the Intro are: observations at high latitudes are rare..., now new capability for day time observations...

Maybe mention also that CALIOP observations are available to monitor SSA as well, but the disadvantage is....

Section 2: ...is much too long. One paragraph and good references would be fine. Section 2 could be even left out..., could be the introductory part of Section 3 (Method).

There are many sentences that must be simply improved: The detection system is capable to detect wavelengths? Simply bad wording... The lidar detects backscatter signals at different wavelengths. There so many, many more examples throughout the paper...., e.g., P5, L5:We use an inelastic counter for the denominator of Eq 2... unbelievable wording. So bad! So low quality of precise thinking! Did any of the co-authors (including the director ...) read the manuscript?

Section 3

Again, the section is too long, and contains many trivial parts. Make it compact, give proper references.

P5, L5: The reference is Raman, 1928! I could not believe what I read! Please provide a proper Raman LIDAR (!) reference here. The same for Rayleigh, 1871, 1899. Please provide a proper Rayleigh lidar reference.

Eq.(3), Eq(4): Please note! Quantities in equations are presented as ONE letter (a, b, c, T , p, that's why we use so often alpha, beta, gamma, ... and lambda, and then with index... if needed). So, please improve Eqs. 3 and 4 accordingly.

P5, L29: ...data is reduced to altitudes above the tropopause... another example of bad wording...

Section 4:

I give up...! ....  only a few remarks : purple drawn profile ... or drawn as a red shade.... Please avoid 'drawn'!... In many cases, you can leave it simply out, sometimes one may use: ... is shown as purple curve, or given as red profile etc...

Section 5

To show the performance of the new procedure.....

So, this new procedure should be already briefly explained in the Intro section.

---

## Author Comment (AC1) · 7 Jun 2019

[a4paper,10pt]article [utf8x]inputenc xcolor

(Author responses are in blue. In the tracked changes version deleted sequences are marked red. New text is marked in blue.) General Comment: We want to thank the three reviewers for the detailed reviews with many useful ideas and suggestions which, we think, have significantly increased the quality of the manuscript. We have rewritten a substantial portion of the manuscript. We restructured the outline of the manuscript. Section 2, formerly named "ALOMAR RMR Lidar" is now called "Instrument and Method" with subsections 2.1 "Processing of the raw data", 2.2 "Calculation

of backscatter ratios" and 2.3 "Identification of the stratospheric aerosol layer". Section 3, formerly named "Methodology" is now named "Calculating the backscatter ratio under daytime conditions". Section 4 contains the results of the paper. A "Summary and Conclusion" can be found in section 5. The nomenclature for the calculation of the backscatter ratio and the color ratio was changed. Therefore, sections 2.2 and 3 have been completely rewritten. The figures have been updated to account for the new symbols.

In the paper, "Year-round stratospheric aerosol backscatter ratios calculated from lidar measurements above Northern Norway", the authors present a multiyear stratospheric sulfate aerosol (SSA) dataset from lidar observations at the ALOMAR research station. This paper provides valuable insight into lidar-measured SSA over the Arctic, and the study is appropriate for AMT, however I have a few major concerns with the paper in its current form. These include the overall writing quality of the manuscript and lack of important details of the study. Thus, I recommend a major revision. The authors should address the major and minor comments outlined below for the revised manuscript.

1. Writing quality of manuscript: Many grammatical errors and misspellings are found throughout the text, and acronyms need to be defined. The paper should be thoroughly proofread.

The paper has been reworked completely to improve the writing quality.
2.Lack of study details: There are several instances in the manuscript that I believe need additional information, as follows:

  a. Page 4, Lines 7-10: Please add more description of ECMWF (e.g., spatial resolution). Why ECMWF? Are there other options? What are the uncertainties associated with the parameters from ECMWF?

  The section has been rewritten. We have selected the ECMWF model as it provides density and ozone data with a time resolution of 1 hour for the location of ALOMAR. We have briefly discussed the use of ozone values from another model

in the manuscript (Page5, line 16).

When you state "converted to a 5 min and 150 m grid", converted from what? Also, add more details on how the Rayleigh and ozone corrections are done.

We interpolate the model data and the lidar data to a grid with a 5 minute time and 150 m vertical spacing. A special bullet point "Gridding of lidar data" has been added to section 2.1

b. Sensitivity studies: Please comment on the choices made, and any sensitivity studies completed for normalization altitudes (Page 5, Line 11), wavelengths for elastic/inelastic signals (Page 5, Lines 22-23), and lower limit of data availability (Page 7, Lines 13-14).

Normalization altitude: The approach was to use the highest possible altitude range as limited by the signal to noise ratio of the Raman backscattered light. This can be seen in Fig 2c: The Signal $S_387$ becomes exceedingly noisy at about 40 km, thus we used a range below. First, we used a range of 30-34 km (as suggested by previous publications) but the results showed, that the upper boundary of the aerosol layer was found above 30 km in many cases. Thus we have lifted the normalization altitude. We have improved section 2.2 accordingly.

Wavelengths: All elastic/inelastic wavelength combinations have been analyzed and lead to proper results. We focused on $R^1064/387$ because of lowest effects due to Ozone extinction and highest backscatter ratios. This is now discussed in the revised manuscript.

Lower limit of data availability: We have improved the section 4 accordingly.

c. Equation 1 (Page 3): Where is this from (reference), or how was it determined?

The equation has been corrected. We have added a reference in the manuscript (Kovalev et al, 2004, "'Elastic Lidar: Theory, Practice, and Analysis Methods"', Page 138).

d. Page 6 (Section 4): Please add more discussion/explanation for this section, and the importance/purpose of each figure (Figures 4 through 7). For example, why are you showing $R355/387$ for Figure 5 instead of R at other wavelengths?

The section has been reworked completely. The correction is presented in more detail.

Also, please state why a correction is needed for $CR1064/355$.

The correction is now explained in detail in the discussion of equations 8 to 11.

e. Page 7 (Figures 9 and 10): Explain how these figures were created (e.g., averaging), as was included in the figure captions.

We have updated the manuscript accordingly: We first calculated hourly averaged backscatter ratios smoothed in altitude with a running mean of 1.1 km. Then we calculated the average for the two telescopes. Finally the mean of the hourly profiles is calculated for each month.

3. Conclusions: I believe this entire section needs to be re-worked. Please address the following:

a. Re-define all acronyms.

b. I recommend not referencing figures in this section.

c. Please do not state results that have not been already discussed earlier in the paper. For example, the uncertainties stated in Line 10 of Page 8. This belongs in the Results section.

d. As mentioned above there are grammatical errors in this section.

e. The narrative does not flow well (including ending with a lone sentence), so I recommend re-writing the entire section.

f. I suggest including bullets or something similar to summarize the main findings of the study.

The section has been reworked completely to account for all the comments.

Minor comments: 1. Page 1, Lines 1-12: Please add a few sentences to the abstract describing the primary results of the study.

Done. Abstract has been reworked

2. Page 1, Line 5: Define ALOMAR.

Done

3. Page 1, Line 15: Define SSA. All acronyms should be defined at their first use in the paper.

Done

4. Introduction section: State the location and dates of the study.

Done

5. Page 2, Lines 31-33: lidar measurements of what? R and CR? Explain the parameters of interest. Also, add more motivation as to why this study is important. What is being accomplished/what is the general purpose of this paper?

6. Page 3, Line 5: Add the elevation of the ALOMAR station.

Done

7. Pages 3 and 4 (Section 2): I suggest not using dashes when listing the processing steps. Bullets may work better.

Done

8. Page 4, Line 14: How was this relative uncertainty computed? Please add an explanation to the text.

This section was rephrased to provide the explanation in the manuscript.

9. Page 6, Lines 1-2: The layers are also not associated with PSCs because of the PSC screening metrics described on Page 5, correct?

Correct. We added this information now in the manuscript.

10. Page 7, Line 20: Rephrase "The first picture".

Done

11. Page 7, Line 31: Significantly lower altitudes? Are you comparing 12-18 km to 12-22 km? If so, this sentence does not make sense. This paragraph is confusing, so I recommend revising it.

The paragraph was split, because 2 different effects are discussed. The altitude ranges where not meant to be discussed together. This was made clearer.

12. Page 8, Lines 1-4: How do these findings compare with other studies?

We have included a brief comparison to previous studies in section 4.

13. Figure 2: Add labels, like a-d, to the plots, and refer to them in the caption. How was the altitude range of the stratospheric aerosol layer determined? Also, as a general comment, mention whether the altitudes are referenced to above ground level (AGL) or above mean sea level (AMSL). This should be stated in the text of the paper as well.

We added labels and used them in caption. The altitude range of the aerosol layer in this figure is 15 to 34 km and indicates the altitude range between a high tropopause and the lower boundary of the normalization altitude. All altitudes are referenced to AMSL. This is now stated in the revised manuscript.

14. Figure 4: For the x-axis, I suggest not using a slash symbol (/) here, as this could be confusing. Maybe use "or" instead. Also, the colored shaded areas representing the measurement uncertainties are very difficult to see.

We now use "or" as suggested. The uncertainties are pretty small and therefore hard to see. However they become visible above about 28 km.
15. Figure 5: Please mention in the caption what the shaded area in blue and black vertical line at R355/387 = 1 represent.

The figure has been reworked to make the discussion clearer. We have changed the line color of the black vertical line to gray as this line is just drawn for reference.

16. Figure 8 caption: I suggest re-wording "Time of available data".

Changed to "'Available data in hours'".

17. Figures 1-10: I suggest making the text larger for both the axes and color bars.

Done. All figures have been reworked.

---

## Author Comment (AC2) · 7 Jun 2019

[a4paper,10pt]article [utf8x]inputenc xcolor

(Author responses are in blue. In the tracked changes version deleted sequences are marked red. New text is marked in blue.) General Comment: We want to thank the three reviewers for the detailed reviews with many useful ideas and suggestions which, we think, have significantly increased the quality of the manuscript. We have rewritten a substantial portion of the manuscript. We restructured the outline of the manuscript. Section 2, formerly named "ALOMAR RMR Lidar" is now called "Instrument and Method" with subsections 2.1 "Processing of the raw data", 2.2 "Calculation

of backscatter ratios" and 2.3 "Identification of the stratospheric aerosol layer". Section 3, formerly named "Methodology" is now named "Calculating the backscatter ratio under daytime conditions". Section 4 contains the results of the paper. A "Summary and Conclusion" can be found in section 5. The nomenclature for the calculation of the backscatter ratio and the color ratio was changed. Therefore, sections 2.2 and 3 have been completely rewritten. The figures have been updated to account for the new symbols.

This paper presents the analysis of stratospheric aerosols observations using the state of the art Rayleigh-Mie-Raman multiple wavelength lidar at ALOMAR. The stratospheric aerosol layer is observed at 1064 nm with unprecedented high resolution. The topic of the paper is well suitable for publication in AMT. However the description of the data processing and the results should be improved and I recommend a major revision as detailed below.

Equation (1) page 3 for the dead time correction does not seem correct. The correct formulation is: N=Ncount/(1-tau Ncount)

Corrected

Page 6, lines 10-11, how equation 3 could be applied if the inelastic signal is not present?

We have used the inelastic signal as measured during the night. This was made clearer in section 2.3.

Page 6, lines 15-21, the justification for a linear correction with altitude of the R355/387 is not given. It may hide some instrumental problems in the lidar. This point should be discussed in more details.

A new paragraph was added to cover this comment in section 3: "First of all we have not identified an instrumental problem that leads to this linear decrease with altitude; for example an incomplete overlap function would affect both signals S355 and S387

in the same way. Furthermore Ozone extinction can be excluded as potential source of error as it has virtually no impact on these signals."

Also I wonder why the ratios R532/387 and R532/355 on Figure 4 fall below 1 in the lower altitude range. Is it a problem of detector saturation?

An explanation for this issue is now given in the manuscript: "'$R^532_387$ is affected by ozone extinction. By definition a backscatter ratio should not be smaller than 1. This indicates that the true ozone extinction may be different from that used for processing the data since the signal at $\lambda = 532$ nm is stronger affected by ozone extinction than the signal at $\lambda = 355$ nm. Due to the normalization of the backscatter ratio to 1 in the aerosol free altitude $z_F$ an under-estimation of ozone extinction reduces the backscatter ratio and may result in $R < 1$. A similar effect arises due to a wavelength dependent extinction of the aerosol layer. Here R is reduced at lower altitudes if the wavelength of the elastic backscattered signal is more affected by aerosols than the Raman wavelength."'

Results section page 7 How the standard error of the monthly mean scattering ratio is computed? Is it from the statistical error on lidar signal at different wavelengths? Due to the limited number of available hours of measurements per month and the large variability of the Artcic stratosphere, especially during winter months, the monthly averaged value of the scattering ratio cannot be considered as fully representative of the monthly climatological value for this month.

The standard error of the monthly mean backscatter ratio is computed like follows: $\sigma_m(R) = \sigma/sqrt(n)$ with sigma being the standard deviation of the backscatter ratio for each month and n being the measurement time in hours for each month. This information is now included in the revised manuscript.

Page 7, lines 27-29, the increase of aerosol loading in the lower stratosphere in August-September due to smoke from the Canadian fires merits to be discussed in more details than just put in the mean seasonal cycle.

[Figure]

A reference to a detailed discussion of the event is now given in the revised version. We mention the wildfires just shortly as a confirmation for the reasonable results of the aerosol retrieval.

---

## Author Comment (AC3) · 7 Jun 2019

[a4paper,10pt]article [utf8x]inputenc xcolor

(Author responses are in blue. In the tracked changes version deleted sequences are marked red. New text is marked in blue.) General Comment: We want to thank the three reviewers for the detailed reviews with many useful ideas and suggestions which, we think, have significantly increased the quality of the manuscript. We have rewritten a substantial portion of the manuscript. We restructured the outline of the manuscript. Section 2, formerly named "ALOMAR RMR Lidar" is now called "Instrument and Method" with subsections 2.1 "Processing of the raw data", 2.2 "Calculation

of backscatter ratios" and 2.3 "Identification of the stratospheric aerosol layer". Section 3, formerly named "Methodology" is now named "Calculating the backscatter ratio under daytime conditions". Section 4 contains the results of the paper. A "Summary and Conclusion" can be found in section 5. The nomenclature for the calculation of the backscatter ratio and the color ratio was changed. Therefore, sections 2.2 and 3 have been completely rewritten. The figures have been updated to account for the new symbols.

The paper is appropriate for AMT, but not in a good shape. Major revisions are needed. The paper is much too long. Basic lidar stuff is unnecessarily presented in large detail. A compact version is needed.

Introduction: The importance of the SSA is presented in large detail! Why? One paragraph would be sufficient! On the other hand, one has to read the entire paper to get an idea: What is new here? What is the motivation to write this paper? Figures 9 and 10 tell the reader finally what the step forward is.

The manuscript was rewritten in large parts. We think that the novelty of the method and the motivation for the paper is now clear.

Please provide the motivation right in the beginning (second paragraph of the introduction): precise and compact. The shorter the introduction the better.

We now mention the motivation at the end of the introduction. We tried to shorten the introduction and also tried to take the other referees comment into account.

Maybe mention also that CALIOP observations are available to monitor SSA as well, but the disadvantage is. . ..

We did not include more discussion here as CALIOP does not provide stratospheric backscatter ratios at 1064 nm (Vernier et al., JGR, 2009). A detailed comparison of CALIOP and ground based lidar is given in Khaykin, ACP, 2017.

Section 2: . . .is much too long. One paragraph and good references would be fine.

Section 2 could be even left out..., could be the introductory part of Section 3 (Method).

As described in the general comment, the section has been completely rearranged and also shortened.

There are many sentences that must be simply improved: The detection system is capable to detect wavelengths? Simply bad wording. . . The lidar detects backscatter signals at different wavelengths. There so many, many more examples throughout the paper. . .. , e.g., P5, L5:We use an inelastic counter for the denominator of Eq 2. . . unbelievable wording. So bad! So low quality of precise thinking! Did any of the co-authors (including the director . . .) read the manuscript?

The whole manuscript has been revised.

P5, L5: The reference is Raman, 1928! I could not believe what I read! Please provide a proper Raman LIDAR (!) reference here. The same for Rayleigh, 1871, 1899. Please provide a proper Rayleigh lidar reference.

The section "Calculation of backscatter ratios" has been reworked completely.

Eq.(3), Eq(4): Please note! Quantities in equations are presented as ONE letter (a, b, c, T , p, that's why we use so often alpha, beta, gamma, ... and lambda, and then with index. . . if needed). So, please improve Eqs. 3 and 4 accordingly.

Done. The whole nomenclature for the derivation of the backscatter ratio was reworked.

P5, L29: . . .data is reduced to altitudes above the tropopause. . . another example of bad wording. . .

Rephrased

Section 4: I give up. . .! . . .. only a few remarks : purple drawn profile . . . or drawn as a red shade. . .. Please avoid 'drawn'!... In many cases, you can leave it simply out, some- times one may use: . . . is shown as purple curve, or given as red profile etc. . .

The section "Results" has been reworked completely.

So, this new procedure should be already briefly explained in the Intro section.

Done